# Interplay between UNG and AID governs intratumoral heterogeneity in mature B cell lymphoma

Pilar Delgado[1¤a], Ángel F. Álvarez-Prado[1¤b], Ester Marina-Zárate[1], Isora V. Sernandez[1¤c], Sonia M. Mur[1], Jorge de la Barrera[2], Fátima Sanchez-Cabo[2], Marta Cañamero[3], Antonio de Molina[4], Laura Belver[1¤d], Virginia G. de Yébenes[1¤e], Almudena R. Ramiro[1] *

**1** B Lymphocyte Biology Lab. Centro Nacional de Investigaciones Cardiovasculares (CNIC), Madrid, Spain, **2** Bioinformatics Unit. Centro Nacional de Investigaciones Cardiovasculares (CNIC), Madrid, Spain, **3** Roche Pharma, Penzberg, Germany, **4** Comparative Medicine Unit, Centro Nacional de Investigaciones Cardiovasculares (CNIC), Madrid, Spain

¤a Current address: Centro de Biología Molecular Severo Ochoa, Madrid, Spain
¤b Current address: University of Lausanne and Ludwig Institute for Cancer Research, Lausanne Branch, Epalinges, Switzerland
¤c Current address: Takeda Farmacéutica España, Madrid, Spain
¤d Current address: Josep Carreras Leukemia Research Institute, Barcelona, Spain
¤e Current address: Complutense University School of Medicine, Dept. of Immunology, Ophthalmology and ENT, 12 de Octubre Health Research Institute (imas12), Madrid, Spain
* aramiro@cnic.es

**Data Availability Statement:** Whole Exome Sequencing data are deposited at https://www.ncbi.nlm.nih.gov/bioproject/PRJNA541807 under project ID: PRJNA541807.

## Abstract

Most B cell lymphomas originate from B cells that have germinal center (GC) experience and bear chromosome translocations and numerous point mutations. GC B cells remodel their immunoglobulin (Ig) genes by somatic hypermutation (SHM) and class switch recombination (CSR) in their Ig genes. Activation Induced Deaminase (AID) initiates CSR and SHM by generating U:G mismatches on Ig DNA that can then be processed by Uracyl-N-glycosylase (UNG). AID promotes collateral damage in the form of chromosome translocations and off-target SHM, however, the exact contribution of AID activity to lymphoma generation and progression is not completely understood. Here we show using a conditional knock-in strategy that AID supra-activity alone is not sufficient to generate B cell transformation. In contrast, in the absence of UNG, AID supra-expression increases SHM and promotes lymphoma. Whole exome sequencing revealed that AID heavily contributes to lymphoma SHM, promoting subclonal variability and a wider range of oncogenic variants. Thus, our data provide direct evidence that UNG is a brake to AID-induced intratumoral heterogeneity and evolution of B cell lymphoma.

## Author summary

Activation Induced Deaminase (AID) is a critical enzyme of the immune response because it gives rise to higher affinity antibodies as well as to various antibody isotypes

**Funding:** PD was supported by the AECC foundation (AIO2012). EM-Z is an FPI fellow (BES-2014-069525). AFA-P, S.M.M. and A.R.R. are supported by CNIC funding. This project was funded by the Spanish Ministerio de Economía, Industria y Competitividad SAF2013-42767-R, SAF2016-75511-R, and European Research Council StG BCLYM, to A.R.R. The CNIC is supported by the Ministerio de Ciencia, Innovacion y Universidades and the Pro-CNIC Foundation, and is a Severo Ochoa Center of Excellence (SEV-2015-0505). The funders had no role in study design, data collection and analysis, decision to publish, or preparation of the manuscript.

**Competing interests:** The authors have declared that no competing interests exist.

(IgG, IgA, IgE). AID introduces deaminations (a pre-mutagenic alteration in the DNA) on the antibody genes, which are then processed by uracil-N-glycosylase (UNG). Occasionally, AID can target other genes, resulting in potentially oncogenic mutations or chromosome translocations. Here we have explored the contribution of AID and UNG to lymphoma generation by combining gain- and loss-of-function genetic mouse models. We find that increasing the activity of AID does not, by itself, increase the incidence of lymphomas in mice. In contrast, mice lacking UNG activity are more susceptible to suffer lymphoma when AID is expressed at higher levels. By sequencing the exomes of lymphomas from these mice, we have learned that AID can contribute to lymphoma development in a translocation-independent fashion, by increasing the variability of oncogenic mutations present in the tumors. Thus, UNG acts as a brake to AID-induced intratumoral heterogeneity and evolution.

## Introduction

Non-Hodgkin-lymphomas (NHL) are high incidence hematological malignancies, with more than 500,000 new cases diagnosed in 2018 worldwide (World Cancer Research Fund, www.wcrf.org). NHL comprise a number of distinct pathological entities, however, the vast majority of them arise from the malignant transformation of mature B cells with germinal center (GC) experience. Those include follicular lymphoma (FL) and the more aggressive diffuse large B cell lymphoma (DLBCL), which together account for more than 50% of all NHL, as well as Burkitt lymphoma (BL) [1].

The etiological relevance of GCs in the malignant transformation of B cells is complex and multi-layered [2,3]. GCs are microanatomical structures that are formed in secondary lymphoid organs upon B cell stimulation during the immune response. When activated by antigen in the context of an antigen presenting cell and a cognate T cell, B cells engage in an intense proliferative reaction and are subject to point mutations at the variable, antigen recognition portion of their antibody (or immunoglobulin, Ig) genes, a process called somatic hypermutation (SHM). SHM, combined with affinity maturation, allows the generation of higher affinity antibodies. In addition, T-cell-activated B cells undergo a region-specific recombination reaction called class switch recombination (CSR), which replaces the primary IgM/IgD constant region by alternative downstream constant regions at the Ig heavy (IgH) locus, thus giving rise to IgG, IgE or IgA isotypes and adding enormous functional versatility to antibody-mediated antigen clearance [4–8].

CSR and SHM are triggered by a single enzyme called activation induced deaminase (AID) through an initial cytosine to uracil deamination event on the DNA of Ig genes. AID deficiency dramatically blocks SHM and CSR and causes human HIGM2 immunodeficiency [9,10]. AID-mediated deamination results in a U:G mismatch that can then be recognized by the base excision repair (BER) or the mismatch repair (MMR) pathways to ultimately give rise to a mutation (in the case of SHM) or a double strand break and a recombination reaction, in CSR. Accordingly, $Ung^{-/-}$ -BER deficient- mice, and $Msh2^{-/-}$ -MMR deficient- mice show different alterations in SHM and CSR [11–18]. Likewise, double $Ung^{-/-}Msh2^{-/-}$ deficient mice only retain transition mutations at C/G, directly emerging from the replication of U:G mismatches, and are devoid of other SHM footprints, as well as of CSR [19,20].

Early studies showed that several lymphoma-associated oncogenes bore mutations with the hallmark of SHM [21–23], and the molecular contribution of AID activity to off-target SHM was directly shown in mice later [20,24], indicating that a relatively high proportion of genes

can be targets of AID mutagenic activity. Moreover, AID can trigger chromosome transloca-
tions involving *IgH* and *Myc*, the hallmark of Burkitt lymphoma [25–27], as well as many
other genomic locations [28,29]. Notably, AID-dependent *myc/IgH* translocations are abol-
ished in the absence of UNG [25], suggesting that UNG activity is required for the processing
of AID deaminations into the DNA double strand break preceding the chromosome transloca-
tion joining. On the other hand, SHM frequency is increased in the combined absence of
UNG and MSH2, indicating that BER and MMR contribute together to the faithful repair of
AID-induced deaminations [4,20,24,30]. In line with these observations, UNG deficient mice
develop B cell lymphomas [31], indicating a tumor suppressor role for UNG. However, the
contribution of UNG to lymphomagenesis is intricate and not completely understood. On one
hand, UNG deficiency protects against the development of BCL6-driven DLBCL in IμHABcl6
mice [32]. On the other, UNG protects B cells from telomeric loss mediated by AID [33], and
this can in turn result in a tumor enabling activity by protecting cell fitness [34].

In addition, various studies have correlated AID activity with human B cell neoplasia. AID
expression is associated to bad prognosis in hematological malignancies [35–39] and foot-
prints of SHM have been found in human B cells [21,22,40] and B cell neoplasia [23,41,42]. In
addition, genetic mouse models have been used to explore the causative role for AID in B cell
transformation. Several loss of function approaches have shown that AID deficiency delayed
the appearance of B cell malignancies [26,37,43] and that this effect might be specific of mature
B cell neoplasias [37,44–46]. However, gain-of-function approaches uncovered an unexpected
complexity on the contribution of AID to B cell malignant transformation. In an early study
where AID was expressed as an ubiquitous transgene, T, but not B cell lymphomas were gener-
ated [47]; this idea of tissue- or stage- specific sensitivity to AID-induced transformation was
also proposed in other transgenic models [48]. Two additional AID transgenic mouse models
failed to promote B cell transformation, presumably due to B cell specific negative regulation
of AID levels [49] or activity [50]. Finally, an AID transgene driven by Ig light chain regulatory
regions revealed that although AID promoted genomic instability, B cell transformation
required concomitant loss of the tumor suppressor p53 [51]. Thus, the contribution of AID,
and specifically of SHM, to B cell transformation is not fully understood.

Here, using a non-transgenic conditional mouse model for AID supra-expression, we
found that AID expression alone was not sufficient to promote B cell lymphomagenesis,
regardless of the onset of AID expression and even in the absence of p53. In contrast, removal
of BER activity by breeding to *Ung*$^{-/-}$ mice resulted in an increase of SHM and of B cell lym-
phoma incidence. Molecular characterization of *Ung*$^{-/-}$ tumors by whole exome sequencing
revealed that AID supra-expression increased intratumor heterogeneity (ITH), giving rise to a
more diverse range of oncogenic alterations. Thus, our work establishes formal genetic evi-
dence for the contribution of AID-mediated SHM to B cell lymphoma tumor evolution.

## Results

### A mouse model to address the contribution of AID to B cell lymphomagenesis

To address the contribution of AID activity to B cell lymphomagenesis we made use of a con-
ditional mouse model for AID expression [52]. A cassette containing the *Aicda* and *EGFP*
cDNAs separated by an internal ribosomal entry site (IRES) was placed into the endogenous
*Rosa26* locus preceded by a floxed transcriptional stop sequence (*R26*$^{+/AID}$ mice, S1A Fig,
[52]). Thus, AID expression from the *Rosa26-Aicda* allele is silent until the transcriptional stop
is removed by Cre. To approach the role of AID in B cell lymphomagenesis we induced AID
expression in B cells by breeding *R26*$^{+/AID}$ mice to *Cd19*$^{+/cre}$ mice, where *Cre* is inserted in the

B-cell specific *Cd19* allele [53] ($R26^{+/AID}Cd19^{+/cre}$ mice). We observed that the onset of expression from the *Rosa26-Aicda* allele occurred early in B cell precursors and progressively increased during B cell differentiation, being almost complete in mature B cells, and consistently with previous reports using the *Cd19-Cre* strain (S1B Fig) [54,55]. Expression of AID at early stages did not detectably affect bone marrow differentiation (S1C Fig).

To assess AID expression in mature B cells from $R26^{+/AID}Cd19^{+/cre}$ mice, we performed a qRT-PCR analysis that allowed quantification of AID mRNA expressed specifically from the *Rosa26-Aicda* allele, or total AID mRNA, resulting from the combined expression from the endogenous AID allele and the *Rosa26-Aicda* allele. Expectedly, we detected expression of *Rosa26-Aicda* both in naïve and activated B cells from $R26^{+/AID}Cd19^{+/cre}$ mice and not in B cells from control mice (Fig 1A, top graphs). Accordingly, assessment of total AID mRNA showed that the *Rosa26-Aicda* allele promotes expression of AID in naïve B cells (Fig 1A, bottom left) and significantly increases the total expression level of AID in activated B cells (Fig 1A, bottom right). In agreement with this result, western blot analysis in naïve B cells revealed AID protein only in $R26^{+/AID}Cd19^{+/cre}$ mice, while LPS+IL4 activated B cells had increased AID protein levels in $R26^{+/AID}Cd19^{+/cre}$ mice (1.6 fold increase), compared to $R26^{+/+}Cd19^{+/cre}$ control mice (Fig 1B and 1C). Thus, we have generated a model for expression of AID in naïve B cells and supra-expression in activated B cells.

To assess the activity of the *Rosa26-Aicda* allele we first performed *in vitro* stimulations of spleen naïve B cells in the presence of LPS and IL4 and found that the frequency of CSR to IgG1 was increased by roughly 1.5 fold in $R26^{+/AID}Cd19^{+/cre}$ B cells when compared to $R26^{+/+}Cd19^{+/cre}$ control B cells (Fig 1D). We next assessed GC formation and CSR *in vivo* after mouse immunization with sheep red blood cells (SRBC). We found that $R26^{+/AID}Cd19^{+/cre}$ mice accumulated more IgG1+ switched cells than did $R26^{+/+}Cd19^{+/cre}$ cells (1.5 fold increase), while the proportion of Fas+ GL7+ GC B cells was not affected (Fig 1E and 1F). We conclude that $R26^{+/AID}Cd19^{+/cre}$ mice display increased AID expression and activity.

## Early AID supra-expression is not sufficient to promote B cell lymphoma

To evaluate the contribution of AID supra-expression to B cell lymphomagenesis, we generated cohorts of $R26^{+/+}Cd19^{+/cre}$ and $R26^{+/AID}Cd19^{+/cre}$ mice for ageing experiments. Mice were inspected weekly and animals showing symptoms of overt disease were euthanized. We found that survival was not affected in $R26^{+/AID}Cd19^{+/cre}$ compared to $R26^{+/+}Cd19^{+/cre}$ mice (Fig 2A, left) and that lymphoma incidence was not increased in $R26^{+/AID}Cd19^{+/cre}$ mice (Fig 2A, right, and not shown). Cre-driven expression of AID in $R26^{+/AID}Cd19^{+/cre}$ mice occurs progressively and relatively late during B cell differentiation. To assess whether the lack of oncogenic impact of AID in $R26^{+/AID}Cd19^{+/cre}$ mice was due to a specific resistance of mature B cells to AID oncogenic transformation, we generated $R26^{+/AID}Vav-Cre^{+/TG}$ mice, where AID supra-expression is driven by *Vav-Cre* from very early hematopoietic progenitors [56]. Reporter GFP expression was detected from early lymphoid development as expected (S2 Fig). We again found no survival reduction and no apparent histological differences in $R26^{+/AID}Vav-Cre^{+/TG}$ mice compared to control $R26^{+/+}Vav-Cre^{+/TG}$ mice (Fig 2B). Likewise, $R26^{+/AID}mb1^{+/cre}$ mice did not show increased lymphoma incidence compared to $R26^{+/+}mb1^{+/cre}$ littermates [46].

It has previously been shown in a different mouse model that deregulated expression of AID is insufficient to induce B cell lymphoma and that concomitant loss of p53 is required for transformation [51]. To address whether in our $R26^{+/AID}Cd19^{+/cre}$ mice p53 tumor suppressor activity was preventing AID-induced oncogenic transformation, we generated $R26^{+/AID}Cd19^{+/cre}Tp53^{-/-}$ mice. We found that survival was not affected in $R26^{+/AID}Cd19^{+/cre}Tp53^{-/-}$ mice when compared to $R26^{+/+}Cd19^{+/cre}Tp53^{-/-}$ (Fig 2C, left). Likewise, the majority of tumor-bearing mice succumbed

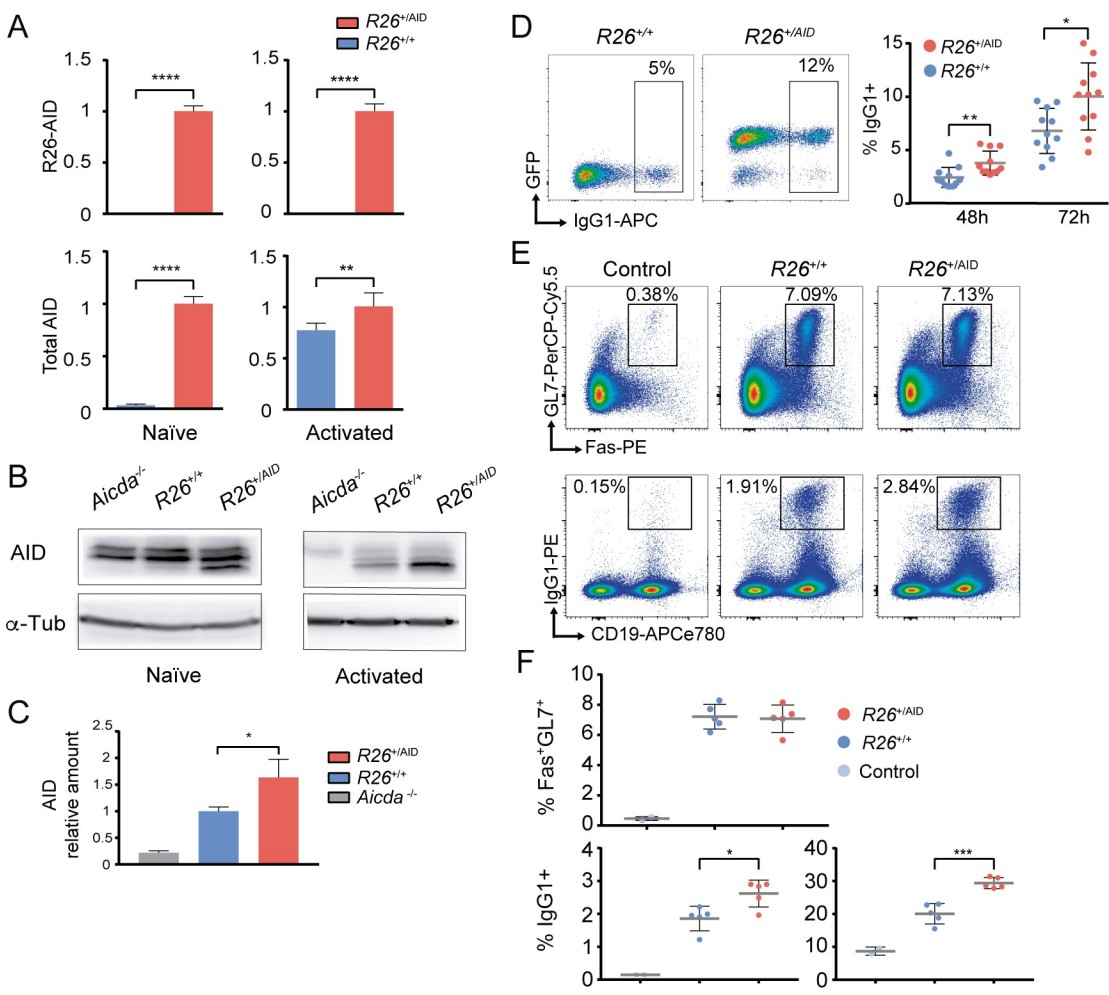

**Fig 1. Characterization of $R26^{+/AID}Cd19^{+/Cre}$ mice.** (**A**) Increased expression of *Aicda* mRNA in B cells from $R26^{+/AID}Cd19^{+/Cre}$ mice. mRNA levels of *R26-AID* and total *Aicda* was measured by RT-qPCR in naïve (left) and in LPS+IL4 activated B (right) cells from $R26^{+/AID}Cd19^{+/Cre}$ (red bars) and $R26^{+/+}Cd19^{+/Cre}$ control mice (blue bars). Values are relative to levels in naïve (left) or activated (right) B cells from $R26^{+/AID}Cd19^{+/Cre}$ mice; mean of two experiments is shown; n = 6 per group. (**B**) Increased AID protein in B cells from $R26^{+/AID}Cd19^{+/Cre}$ mice. Total lysates of naïve or LPS+IL4 activated B cells from $R26^{+/AID}Cd19^{+/Cre}$ and $R26^{+/+}Cd19^{+/Cre}$ mice were analysed by western blot with a monoclonal anti-AID antibody. Incubation with an anti-tubulin antibody is shown as loading control. (**C**) Quantification of AID protein in LPS+IL4 activated B cells by western blot. Total lysates of activated B cells from 4 individual mice of each genotype, $R26^{+/AID}Cd19^{+/Cre}$ and $R26^{+/+}Cd19^{+/Cre}$ and 1 *Aicda*$^{-/-}$ were analyzed by western blot as shown in (B). Values are relative to $R26^{+/+}Cd19^{+/Cre}$ and normalized to actin or tubulin. (**D**) Increased CSR in $R26^{+/AID}Cd19^{+/Cre}$ mice. Representative FACS analysis of CSR to IgG1 in LPS+IL4 activated B cells from $R26^{+/AID}Cd19^{+/Cre}$ and $R26^{+/+}Cd19^{+/Cre}$ mice (left panel). Quantification is shown on the right. Each dot represents an individual mouse (n = 11 mice per group). (**E**) Increased immunization response in $R26^{+/AID}Cd19^{+/Cre}$ mice. Representative FACS analysis of GC (Fas+GL7+) B cells (upper panels) and CSR to IgG1 (lower panels) in non-immunized $R26^{+/+}Cd19^{+/Cre}$ (control, PBS) and SRBC-immunized $R26^{+/+}Cd19^{+/Cre}$ and $R26^{+/AID}Cd19^{+/Cre}$ mice after B220+ gating. (**F**) Quantification of Fas+GL7+ GC B cells (upper panel) and CSR to IgG1 (lower graphs) on B220+-gated (lower left) and B220+Fas+GL7+-gated B cells (lower right) from the immunization experiment shown in (E). Each dot represents an individual mouse (n = 2–5 mice per group). * p≤0.05; ** p≤0.005; **** p≤0.0001; two-tailed t-test.

to T cell lymphoma or solid tumors both in $R26^{+/+}Cd19^{+/cre}Tp53^{-/-}$ and in $R26^{+/AID}Cd19^{+/cre}Tp53^{-/-}$ mice (65% and 77% combined frequency, respectively).

Thus, supra-physiological expression of AID in R26$^{+/AID}$ mice did not impact on the overall mouse survival or the incidence of B cell lymphomas, regardless of the onset of expression or the absence of p53 tumor suppressor.

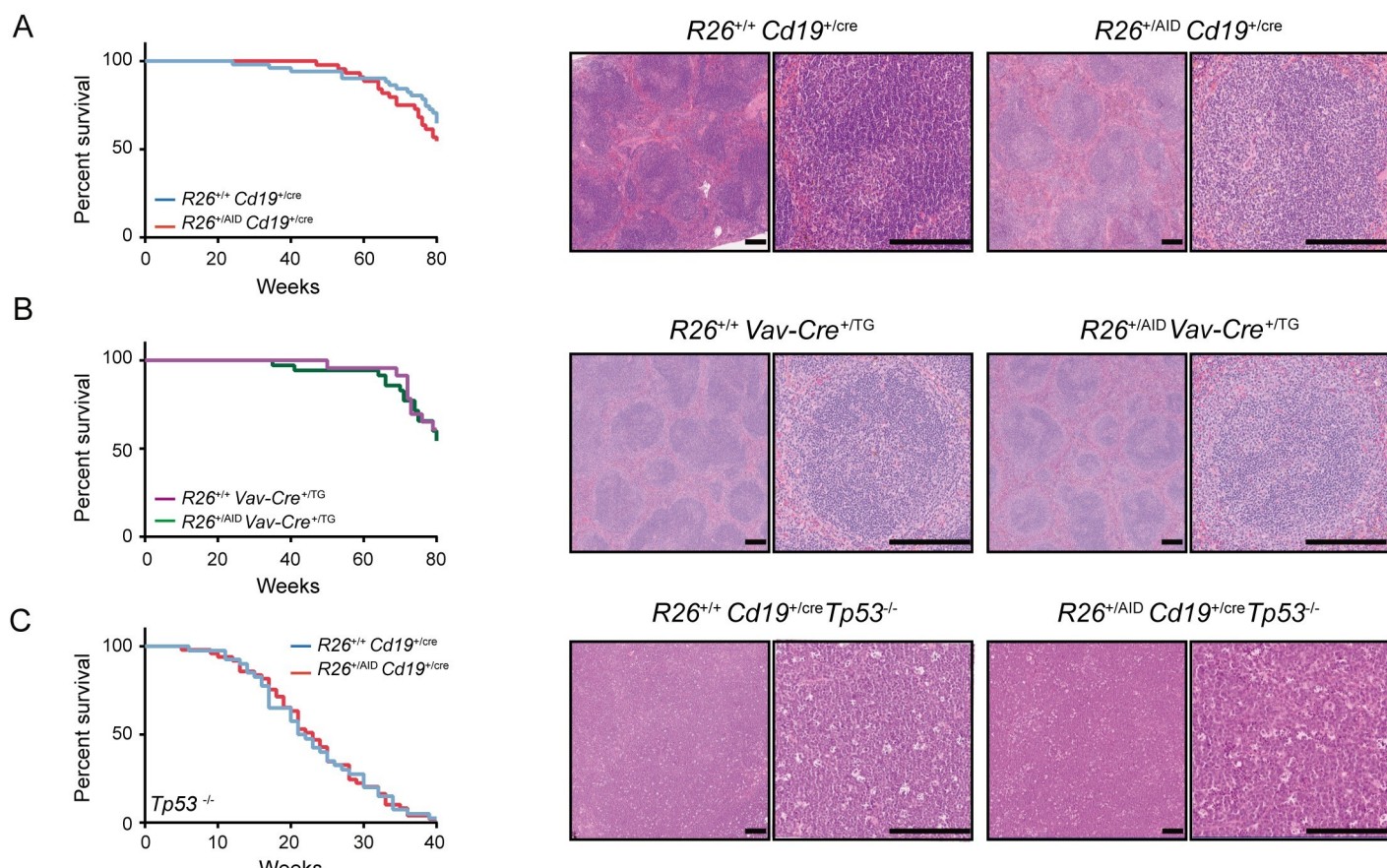

**Fig 2. *R26^{+/AID}Cd19^{+/Cre}* mice do not develop lymphoma.** Kaplan-Meier survival analysis (left) and representative H&E staining of spleen sections (right) of **(A)** *R26^{+/+}Cd19^{+/Cre}* (n = 51) and *R26^{+/AID}Cd19^{+/Cre}* mice (n = 44) mice and **(B)** *R26^{+/+} Vav ^{+/cre}* (n = 23) and *R26^{+/AID} Vav ^{+/cre}* (n = 35) mice. **(C)** Kaplan-Meier survival analysis of *R26^{+/+}Cd19^{+/Cre}Tp53^{-/-}* (n = 40) and *R26^{+/AID}Cd19^{+/Cre}Tp53^{-/-}* (n = 49) and representative H&E staining of thymus T cell lymphomas found in both groups of mice. Mice were followed for 80 weeks (A-B) or at humane endpoint (C). p>0.05 in all cases; log-rank Mantel-Cox test. Magnification is 5x (left) and 20x (right); scale bar is 200μm.

## UNG deficiency increases mutation load in B cells from *R26^{+/AID}Cd19^{+/cre}* mice

We have previously shown that the combined action of BER and MMR can faithfully repair a big fraction of the mutations triggered by AID deamination [20,30]. To determine if under supra-physiological AID expression abolishing BER would result in a higher mutation load, we generated *R26^{+/AID}Cd19^{+/cre} Ung^{-/-}* mice. AID-induced SHM frequency was assessed at the Sμ region of the *IgH* locus by PCRseq [30] in naïve B cells or in GC B cells isolated from SRBC-immunized mice. As controls, we used naïve or GC B cells from *R26^{+/+}Cd19^{+/cre}Ung^{+/-}* mice, *R26^{+/AID}Cd19^{+/cre}Ung^{+/-}* mice and *R26^{+/+}Cd19^{+/cre}Ung^{-/-}* mice. In addition, we included LPS+IL4 activated B cells from AID deficient mice (*Aicda^{-/-}*) as a negative control. We found that in naïve B cells, early AID supra-expression triggered detectable accumulation of mutations only in the absence of UNG (*R26^{+/AID}Cd19^{+/cre} Ung^{-/-}* versus *R26^{+/AID}Cd19^{+/cre}Ung^{+/-}* or *R26^{+/+}Cd19^{+/cre}Ung^{-/-}*) (S1 Appendix, Fig 3A, left). We obtained similar results in GC B cells isolated from SRBC-immunized mice, where mutation load driven by *Rosa26-Aicda* is very much increased when combined with UNG deficiency, but shows only as a trend in an UNG proficient background (Fig 3A, right). Separate analysis of transition and transversion

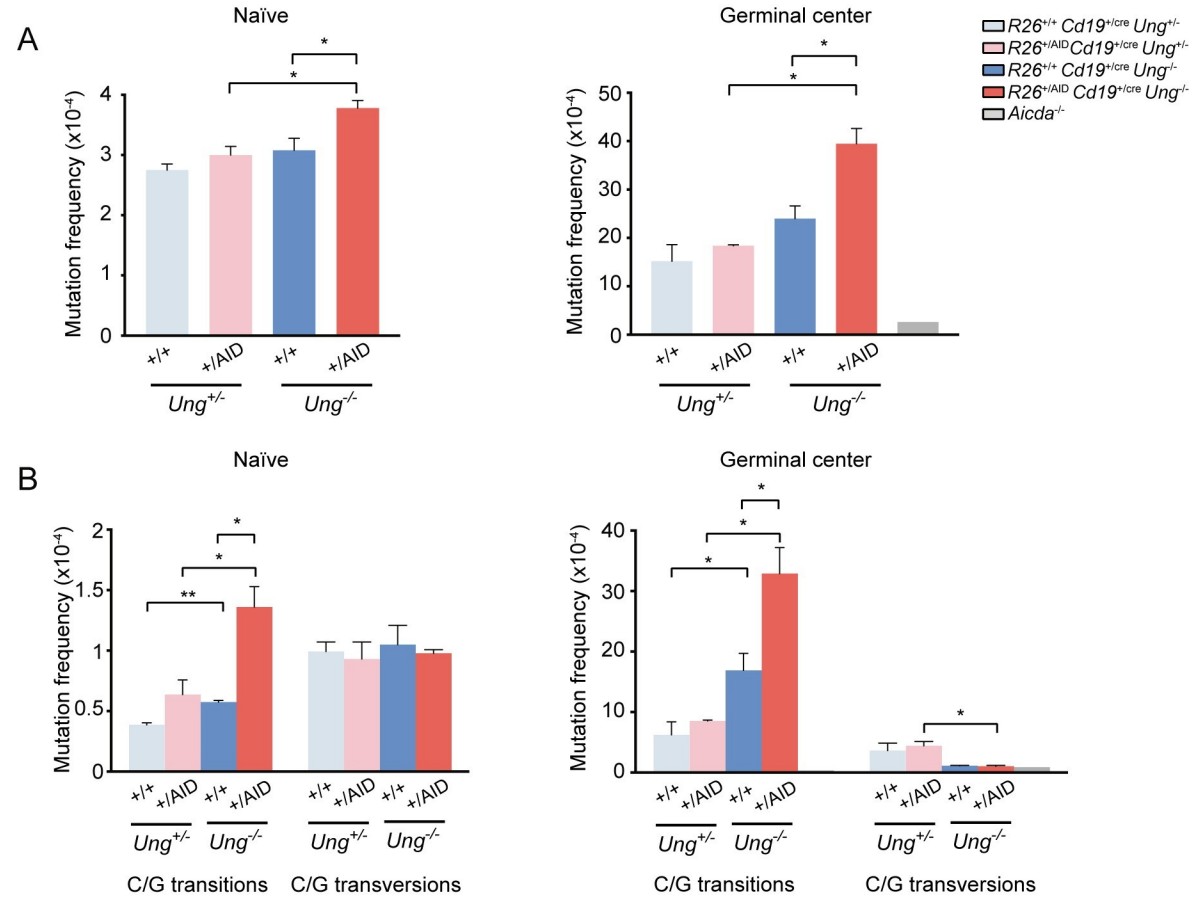

**Fig 3. Increased SHM in BER deficient mice.** Mutation analysis of IgH Sμ region in naïve and GC B cells from $R26^{+/+}$ $Cd19^{+/cre}$ $Ung^{+/-}$, $R26^{+/AID}$ $Cd19^{+/cre}$ $Ung^{+/-}$, $R26^{+/+}$ $Cd19^{+/cre}$ $Ung^{-/-}$, $R26^{+/AID}$ $Cd19^{+/cre}$ $Ung^{-/-}$ and $Aicda^{-/-}$ mice by PCR-Seq. (A) Total mutation frequency and (B) frequency of transition and transversion mutations at C/G lying in WR<u>C</u>Y/RG<u>Y</u>W hotspots. (n = 2, each composed of a pool of 3 mice; * p ≤ 0.05; two-tailed t-test; error bars represent SD).

mutations showed that UNG deficiency promotes an increase at C/G transitions both under endogenous AID and AID supra-expression contexts. Thus, the observed increase in mutation frequency in UNG deficient cells was accounted by an increase in transitions at C/G pairs (Fig 3B), consistent with the outcome of direct replication of unrepaired U:G mismatches [11,15].

We conclude that UNG deficiency increases the mutation load in B cells that supra-express AID. Given that chromosome translocations are impaired in UNG deficient B cells [25], $R26^{+/AID}Cd19^{+/cre}Ung^{-/-}$ mice provide a unique opportunity to assess the contribution of AID-induced SHM to lymphomagenesis.

## UNG deficiency promotes oncogenic transformation by AID

To address the contribution of AID-induced SHM to oncogenic transformation, we generated cohorts of $R26^{+/+}Cd19^{+/cre}Ung^{+/-}$ mice, $R26^{+/AID}Cd19^{+/cre}Ung^{+/-}$ mice and $R26^{+/+}$ $Cd19^{+/cre}$ $Ung^{-/-}$ and $R26^{+/AID}Cd19^{+/cre}Ung^{-/-}$ mice and performed ageing experiments. We found that $R26^{+/AID}Cd19^{+/cre}Ung^{-/-}$ mice showed a reduced median survival compared to the other groups of mice (Fig 4A). Moreover, at sacrifice $R26^{+/AID}Cd19^{+/cre}Ung^{-/-}$ mice harboured a different pattern of B cell lymphomas by immunohistochemistry evaluation, with a smaller proportion of large-B cell (LB) lymphoma and a bigger proportion of small B cell (SB) lymphoma than

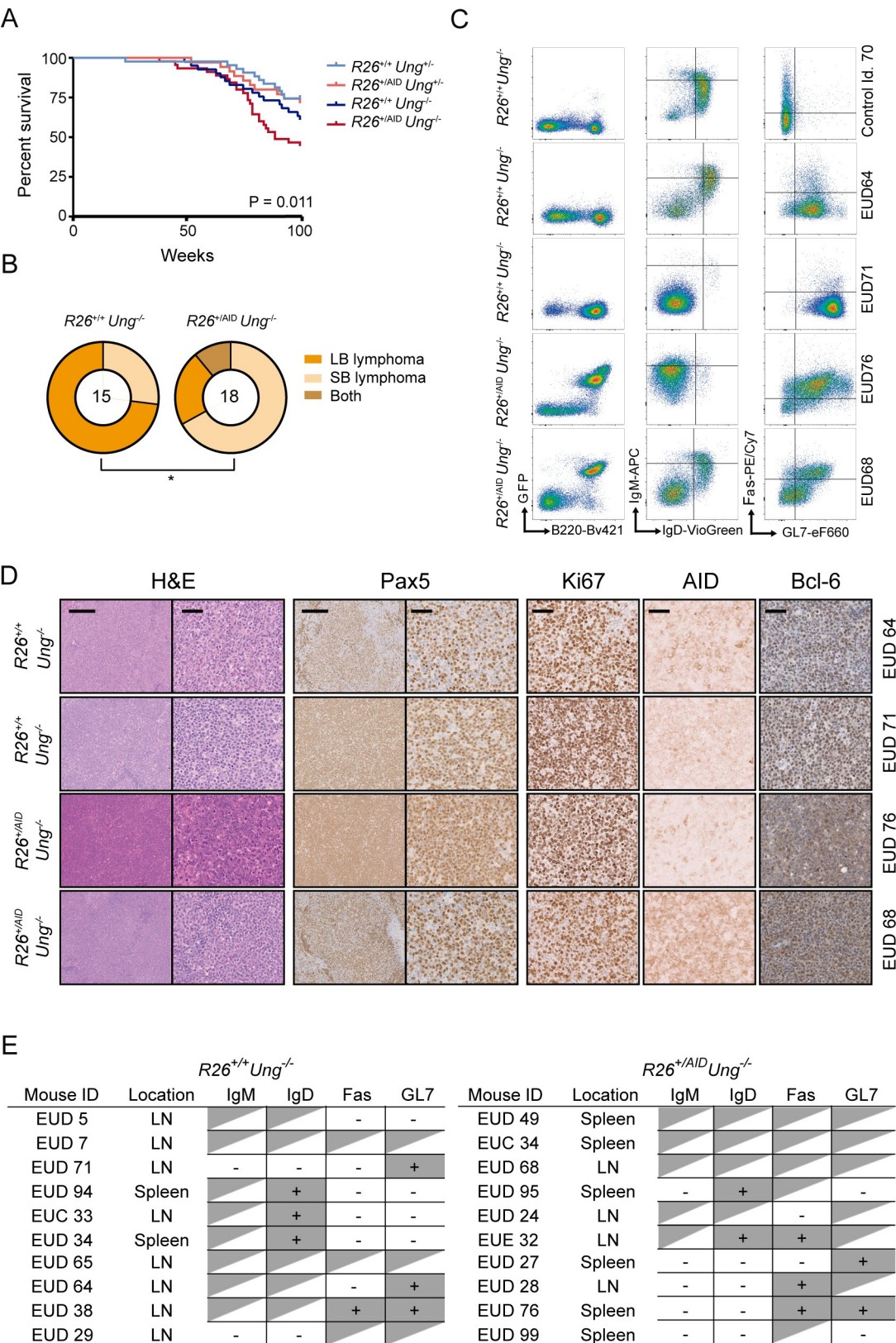

**Fig 4. BER deficiency promotes AID-dependent lymphomagenesis. (A)** Kaplan-Meier survival analysis of $R26^{+/+}Cd19^{+/}$
$^{cre}Ung^{+/-}$(n = 43), $R26^{+/AID}Cd19^{+/cre}Ung^{+/-}$ (n = 35), $R26^{+/+}Cd19^{+/cre}Ung^{-/-}$ (n = 41) and $R26^{+/AID}Cd19^{+/cre}Ung^{-/-}$ mice

(p = 0.011, Log-rank Mantel-Cox test). **(B)** Lymphoma diagnosis of *R26^+/+Cd19^+/cre Ung^-/-* and *R26^+/AID Cd19^+/cre Ung^-/-* mice. Outer rings depict proportion of lymphomas with a SB or LB lymphoma phenotype; inner circle indicates the number of lymphoma tumors analyzed. * p ≤ 0.05, Two-tailed Fisher test. **(C)** Immunophenotype of a control mouse spleen (upper panels) and of representative tumors from *R26^+/+Cd19^+/cre Ung^-/-* and *R26^+/AID Cd19^+/cre Ung^-/-* mice (lower panels). Total live cells (left panel) and B220+ gated cells (middle and right panels) are shown. **(D)** Immunohistochemistry analysis of representative tumors from *R26^+/+Cd19^+/cre Ung^-/-* and *R26^+/AID Cd19^+/cre Ung^-/-* mice. Magnification is 10x for H&E and Pax5 (left panels) and 40x for the rest of the panels. Scale bars are 250μm and 50μm for 10x and 40x images respectively. **(E)** Summary of immunophenotyping markers for a collection of 10 tumors per genotype (as shown in S4 and S5 Figs. Shade (+) indicates positive staining, white indicates negative staining, triangle shade indicates heterogenous staining.

*R26^+/+Cd19^+/cre Ung^-/-* mice (Figs 4B, 4D, S3, S4 and S5; see Methods for details). FACS and immunohistochemistry analysis showed heterogeneity, but tumors from *R26^+/+Cd19^+/cre Ung^-/-* and *R26^+/AID Cd19^+/cre Ung^-/-* mice typically expressed B cell markers, including Pax or B220 as well as the proliferation antigen Ki67 (Figs 4C, 4E, S4 and S5). In addition, we observed that many tumors expressed GL7 and Bcl-6 (Figs 4C, 4E, S4 and S5). Likewise, AID expression was retained in *R26^+/AID Cd19^+/cre Ung^-/-* tumors, but was also common in *R26^+/+Cd19^+/cre Ung^-/-* tumors, supporting a GC origin (Figs 4D, S4 and S5). Together, these results indicate that under the condition of UNG deficiency, AID supra-expression promotes GC B cell transformation.

## AID promotes intratumor heterogeneity

To analyse the contribution of AID to lymphomagenesis at the molecular level, we performed Whole Exome Sequencing (WES) of *R26^+/+Cd19^+/cre Ung^-/-* (n = 7) and *R26^+/AID Cd19^+/cre Ung^-/-* (n = 8) tumors and a healthy tissue control. We sequenced a total of 148Gb with an average depth of 79x (S1 Table and S6 Fig). Variant calling analysis of tumoral versus healthy tissue (see Methods for details) revealed a total number of 4801 unique variants in 5195 different genes, with an average of ~640 variants and ~740 genes affected per tumor (S6A Fig). Most of the variants corresponded to single nucleotide variants (SNVs, ~76%) and a minor fraction were insertions or deletions (INDELS) (~24%) (S6A and S6B Fig).

To assess the mutation load of *R26^+/+Cd19^+/cre Ung^-/-* and *R26^+/AID Cd19^+/cre Ung^-/-* tumors, we first calculated mutation frequencies per genotype. Since in the absence of UNG A/T off-target mutations are minor [20] and transversion mutations at C/G are impaired [11,30] (Fig 3), for this analysis (Fig 5A) only transition mutations at C/G pairs were considered. In *R26^+/+Cd19^+/cre Ung^-/-* tumors, we identified very large numbers of transition mutations at C/G pairs, which were preferentially enriched within WR$\underline{C}$/$\underline{G}$YW (W = A/T; R = C/G; Y = C/T) AID mutational hotspots (Fig 5A), suggesting a contribution of AID to the mutational landscape observed in these tumors. Interestingly, we found a significantly higher mutation frequency and more pronounced hotspot focusing in *R26^+/AID Cd19^+/cre Ung^-/-* than in *R26^+/+Cd19^+/cre Ung^-/-* tumors (Fig 5A). These results indicate that AID-induced mutations occur at larger numbers in *R26^+/AID Cd19^+/cr Ung^-/-* than in *R26^+/+Cd19^+/cre Ung^-/-* tumors, consistently with our findings in non-transformed GC B cells. We found that copy number variations (CNV) and chromosome translocations occurred at similar frequencies in *R26^+/AID Cd19^+/cre Ung^-/-* and *R26^+/+Cd19^+/cre Ung^-/-* tumors (S6C and S6D Fig and S3 and S4 Tables), in agreement with the idea that AID activity is not expected to give rise to double strand breaks in the absence of UNG [11,15,25,57].

Intratumor heterogeneity (ITH) has been related to poor prognosis and impaired response to treatment [58–60]. To assess if AID-induced SHM could contribute to ITH we measured clonal and subclonal variants in *R26^+/+Cd19^+/cre Ung^-/-* and *R26^+/AID Cd19^+/cre Ung^-/-* tumors as a proxy to estimate ITH [61]. In this context, clonal variants refer to early acquired mutations shared by a big proportion of the cancer cells, as reflected by their high variant allele frequency

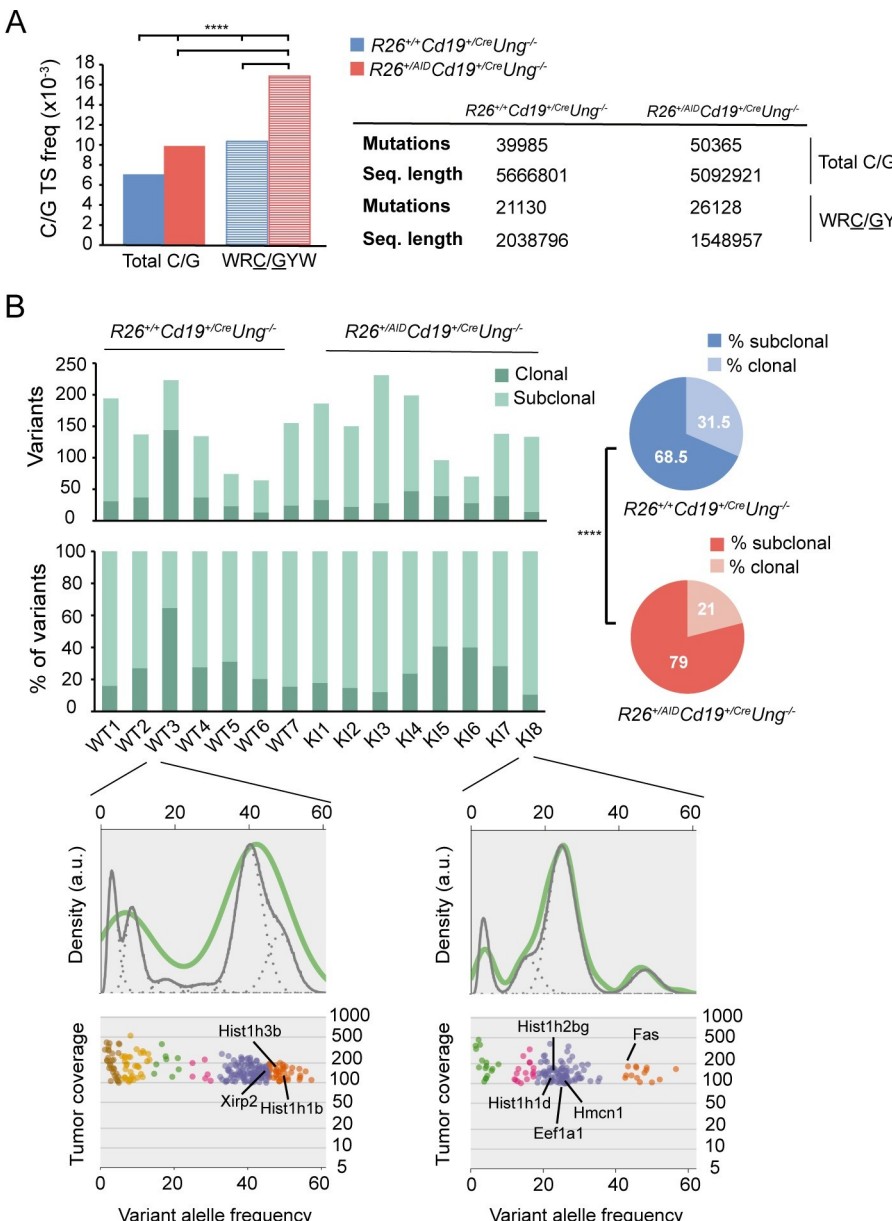

**Fig 5. AID contributes to intratumor heterogeneity. (A)** Average transition mutation frequency at total C/G pairs and at C/G within WR$\underline{C}$/$\underline{G}$YW hotspots in tumors from $R26^{+/+}Cd19^{+/cre}Ung^{-/-}$ and $R26^{+/AID}Cd19^{+/cre}Ung^{-/-}$ mice as measured by whole exome sequencing (Two-tailed $\gamma^2$ test; all comparisons -Total C/G vs WRC/GYW within the same genotype; Total C/G AID WT vs AID KI; WRC/GYW AID WT vs AID KI-, p <2.2x10$^{-16}$). **(B)** Number (upper panel) and proportion (lower panel) of clonal and subclonal variants in each of the seven $R26^{+/+}Cd19^{+/cre}Ung^{-/-}$ (WT) and eight $R26^{+/AID}Cd19^{+/cre}Ung^{-/-}$ (KI) tumors analyzed. Piecharts depict the proportion of clonal and subclonal variants of all the tumors analyzed pooled by genotype (Two-tailed Fisher test; ****p ≤10$^{-4}$). Inset panels show a representative example of the intratumor heterogeneity in each genotype. Clusters as defined by SciClone are depicted in different colors. Clonal variants are those belonging to the cluster with the highest variant allele frequency (VAF; orange clusters in inset panels), usually between 40–50%. Subclonal variants are those belonging to clusters with a lower VAF than that of the clonal cluster. Density plots depict the posterior predictive density, a surrogate of the accuracy of the prediction model assigning variants to the different clusters (green lines represent the diploid model; grey continuous and discontinuous lines represent the model fit of each cluster and each variant, respectively). Tumor coverage refers to the number of reads supporting each variant. Genes that are frequently mutated in human lymphoma tumors are indicated in black.

(VAF). Interestingly, several genes that are frequently mutated in human lymphomas, such as *Fas* [62], *Eef1a1* [63] and some members of the histone family genes (*Hist1h1b*, *Hist1h1d*, *Hist1h2bg*, *Hist1h3b*) [64] bear clonal variants (Fig 5B, inset). Conversely, subclonal variants are mutations arising later in tumor development, defining younger clones with lower VAFs. Thus, a higher proportion of subclonal variants in a tumor reflects a higher ITH. We found that $R26^{+/AID}Cd19^{+/cre}Ung^{-/-}$ tumors have a higher proportion of subclonal variants than $R26^{+/+}Cd19^{+/cre}Ung^{-/-}$ tumors (Fig 5B, right), indicating that AID contributes to the generation of ITH.

To gain insights into the functional relevance of AID activity to the development of lymphoma, we focused on variants affecting known cancer-related genes, including genes that are frequently mutated in human lymphoma [65–71] (see Methods for details). We identified a total number of 132 and 154 cancer-related genes bearing somatic mutations (see Methods) in $R26^{+/+}Cd19^{+/cre}Ung^{-/-}$ and $R26^{+/AID}Cd19^{+/cre}Ung^{-/-}$ tumors, respectively (S2 and S5 Tables; complete list of variants in S3 Table). Notably, a big proportion of these mutated genes are shared between the two groups of tumors (Fig 6A), suggesting a common network of functional pathways being affected for lymphoma generation (Fig 6B and S5 Table). However, we identified several cancer-related genes that were exclusively mutated in $R26^{+/AID}Cd19^{+/cre}Ung^{-/-}$ tumors (Fig 6A and S5 Table). Functional annotation revealed that these genes impinge on more diverse cellular functions, with the p53 DNA damage checkpoint or the negative regulation of B cell activation being more affected in $R26^{+/AID}Cd19^{+/cre}Ung^{-/-}$ than in $R26^{+/+}Cd19^{+/cre}Ung^{-/-}$ tumors (Fig 6B and 6C). Furthermore, mutations in $R26^{+/AID}Cd19^{+/cre}Ung^{-/-}$ exclusive cancer genes occurred more frequently in C/G pairs than in A/T pairs and were enriched in hotspots, strongly suggesting that they have originated from AID activity (Fig 6D). Together, these data indicate that AID induced SHM increases ITH and broadens the range of oncogenic events associated to lymphomagenesis.

## Discussion

Here we have made use of a genetic model for AID expression to assess the contribution of AID to lymphoma development. Our $R26^{+/AID}Cd19^{+/cre}$ model is different from previous approaches in that it is not a transgenic model and it allows conditional expression of AID together with GFP as a positive tracer for AID expression. Expectedly, we find that AID mRNA and protein are detected in naïve B cells in $R26^{+/AID}Cd19^{+/cre}$ and that in activated B cells total AID expression and activity is increased in $R26^{+/AID}Cd19^{+/cre}$ mice compared to $R26^{+/+}Cd19^{+/cre}$ littermates. Thus, in contrast to previous reports [48–50], heterologous *Rosa26-Aicda* expression in $R26^{+/AID}Cd19^{+/cre}$ mice does not seem subject to particular mechanisms of negative regulation, and simply leads to a relatively proportional increase of AID activity.

In agreement with previous data using different mouse models [47–51], we show here that AID supra-expression in $R26^{+/AID}$ mice is not sufficient to promote mature B cell transformation even if driven from very early on in B cell progenitors with *Vav-Cre* or *mb1-Cre* [46] drivers, suggesting that cumulative AID activity during B cell differentiation does not have a measurable oncogenic impact by itself. Interestingly, *Rosa26-Aicda* did not promote mature B cell lymphomas even in the absence of p53, in sharp contrast with a previous report [51]. This discrepancy may well be explained by the difference in AID expression levels achieved in these two studies. While AID expression and activity was only mildly increased in $R26^{+/AID}Cd19^{+/cre}$ mice, it reached much higher levels in the *Igκ*AID mouse from Robbiani et al, when compared to wild type mice, especially in GC B cells. We believe that these expression levels may promote increased genome instability and hence triggered more efficiently a DNA damage response, which would explain the increased incidence of B cell lymphoma observed in the absence of p53 [51].

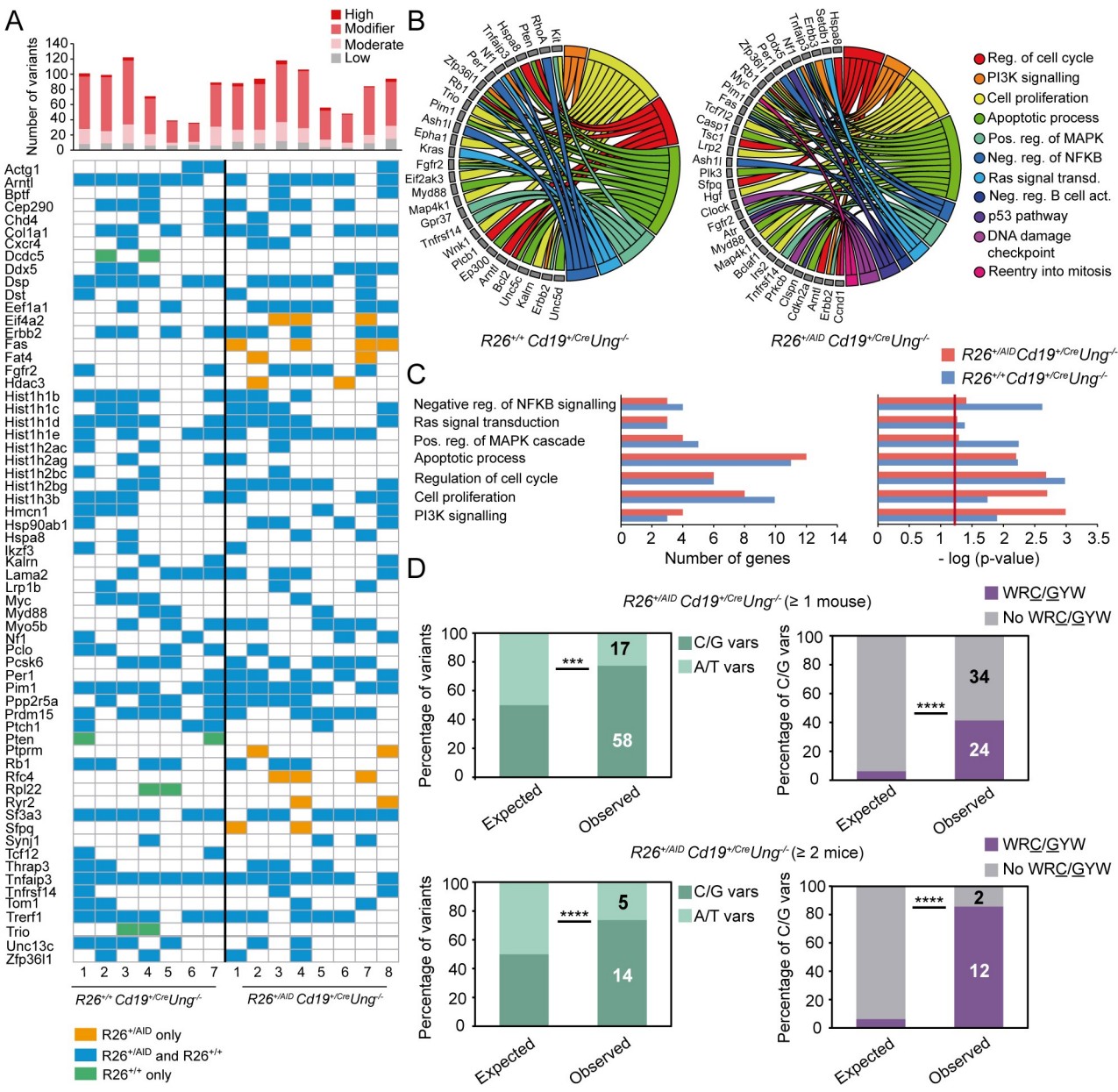

**Fig 6. AID broadens the functional impact of oncogenic events in UNG deficient tumors.** (A) Cancer-related genes found mutated in tumors from $R26^{+/+}Cd19^{+/cre}Ung^{-/-}$ and $R26^{+/AID}Cd19^{+/cre}Ung^{-/-}$ mice. Upper panel shows total number of variants identified in all cancer-related genes in each of the tumors; color code defines the functional impact of the variants as annotated by Variant Effect Predictor (VEP; see methods section for details). Lower panel represents presence (colored box) or absence (empty box) of variants in each of the cancer-related genes for each of the analyzed tumors. For simplicity, only mutated genes found in > = 2 tumors per group are shown; a full list is included in S3 Table. Green indicates genes exclusively mutated in $R26^{+/+}Cd19^{+/cre}Ung^{-/-}$ tumors; orange indicates genes exclusively mutated in $R26^{+/AID}Cd19^{+/cre}Ung^{-/-}$ tumors; blue indicates genes mutated in both. (B) Circos plot representation of the biological functions associated to all mutated cancer-related genes (Gene Ontology biological process database) in $R26^{+/+}Cd19^{+/cre}Ung^{-/-}$ (left) and $R26^{+/AID}Cd19^{+/cre}Ung^{-/-}$ (right) tumors. (C) Number of genes (left) and p-value of the enrichment (right, EASE score -modified Fisher exact test- as provided by DAVID, see methods for details) of the common biological functions affected in both $R26^{+/+}Cd19^{+/cre}Ung^{-/-}$ and $R26^{+/AID}Cd19^{+/cre}Ung^{-/-}$ tumors. (D) Proportion of mutations in C/G and A/T nucleotides (left) and within WR**C**/**G**YW hotspots (right) in cancer-related genes that are exclusively mutated in $R26^{+/AID}Cd19^{+/cre}Ung^{-/-}$ tumors (Fisher two-tailed test; ****$p \leq 10^{-4}$).

In contrast with the lack of lymphomagenic effect of AID, alone or combined with p53 deficiency, we show here that in the absence of UNG, a mild increase in AID expression ($R26^{+/AID}$ $Cd19^{+/cre}$ mice) is sufficient to decrease mean survival and contributes to lymphoma evolution, supporting the notion that UNG can act as a tumor suppressor in the context of B cell lymphomagenesis. Intriguingly, UNG has very recently been shown to improve B cell fitness and to have tumor-enabling activity [34] possibly by protecting B cells from telomeric loss induced by AID [33]. Although seemingly contradictory with our data, it is worth noting that Safavi et al also observed that $Ung^{-/-}$ B cell lymphomas bear the footprint of AID SHM, although the expression of AID was lost in established tumors. Together, these results suggest that the interplay between AID and UNG is a delicate balance where expression levels, and possibly the stage of tumor progression, play a role.

We have observed morphological differences in the lymphomas generated in the presence of AID supra expression, compared to those emerging under endogenous AID expression. However, from this study we cannot ascertain whether they represent a completely different pathological entity or a more aggressive form of the same entity. Likewise, it could well be that early AID supra-expression would favour the transformation of an earlier cell of origin, but this could only be unequivocally tested using specific genetic tracing models.

Regardless of these considerations, our data suggest that increased lymphoma vulnerability of *Ung* deficient mice directly reflects the increased SHM frequency observed when UNG deficiency is combined with *Rosa26-Aicda* expression, while *Rosa26-Aicda* alone (this study) or $Ung^{-/-}$ alone (this study, [20,30]) have only a moderate effect on SHM. Given that UNG is essential for AID-triggered double strand breaks and chromosome translocations, our report provides the first direct genetic proof that the SHM activity of AID can, by itself, contribute to B cell transformation. In turn, the absence of double strand breaks and thus of DNA damage response could favour the survival and accumulation of mutations of GC B cells and their likelihood to oncogenic transformation. Importantly, mutations and homozygous UNG losses have been found, although at low frequency, in human B-NHL[72].

Various studies have previously suggested that AID can drive or contribute to clonal evolution in B cell malignancies [39,73,74]. Our study also provides, to our knowledge, the first causal evidence of AID-induced SHM B cell tumors, as measured by whole exome sequencing in a genetic model. Interestingly, both control $R26^{+/+}Cd19^{+/cre}$ and $R26^{+/AID}Cd19^{+/cre}$ tumors bear the hallmark of AID activity, but this is more pronounced in tumors with supra-expression of AID ($R26^{+/AID}Cd19^{+/cre}$ mice). Further, our study directly shows that AID-induced SHM promotes a bigger proportion of subclones and increased ITH. Our model predicts that AID would increase ITH and contribute to second hit oncogenic transformation events preferentially in tumoral cells with UNG loss-of-function events. Thus, we propose that AID-induced SHM may contribute to lymphoma progression by giving rise to a more diverse output of oncogenic variants that presumably impinge on tumor evolution and aggressiveness [58–60].

## Methods

### Ethics statement

All animals were housed in the Centro Nacional de Investigaciones Cardiovasculares animal facility under specific pathogen-free conditions. All animal procedures conformed to EU Directive 2010/63EU and Recommendation 2007/526/EC regarding the protection of animals used for experimental and other scientific purposes, enforced in Spanish law under RD 53/2013. The procedures have been reviewed by the Institutional Animal Care and Use Committee (IACUC) of Centro Nacional de Investigaciones Cardiovasculares, and approved by

Consejeria de Medio Ambiente, Administración Local y Ordenación del Territorio of Comunidad de Madrid.

## Mice and B cell cultures

$R26^{+/AID}$ mice were described previously [52] and bred to $Cd19^{cre}$ [53], $Vav-Cre^{TG/+}$ [56] $Tp53^{-/-}$ [75] and $Ung^{-/-}$ [76] mice as indicated. $Aicda^{-/-}$ [10] were used as negative control for SHM assays. Littermates aged from 6 to 12 weeks were used for SHM, CSR and immunization experiments.

T-dependent immunization was induced by intravenously injection of $10^8$ sheep red blood cells resuspended in 100 µl of sterile PBS. Immunization response was analyzed in spleen 7 days after sheep red blood cell injection.

For CSR experiments spleen naive B cells were isolated by immunomagnetic depletion using anti-CD43 beads (Miltenyi Biotec). Purified naïve B cells were cultured at 1.2 x $10^6$ cells/ml in complete RPMI medium supplemented with 10% FBS (Gibco), 50 µM 2-βmercaptoethanol (Gibco), 10 mM Hepes (Gibco), penicillin and streptomycin (Gibco), 25 µg/ml lipopolysaccharide (LPS, Sigma-Aldrich) and 10 ng/ml IL-4 (PreproTech).

## Flow cytometry and cell sorting

Single-cell suspensions were obtained from bone marrow, spleen or cultured B cells and stained with fluorochrome/biotin-conjugated antibodies to detect mouse surface antigens: B220 (RA3-6B2), CD19 (1D3), IgM (ll/41), IgD (11-26c.2a), IgG1 (A85-1), CD25 (PC61), Fas (Jo2)), Ly77 (GL7), CD4 (RM4-5), CD8 (53–6.7) and CD44 (IM7). Samples were acquired on LSRFortessa or FACSCanto instruments (BD Biosciences) and analyzed with FlowJo software. GC B cells and naïve B cells were purified from spleen from immunized mice using FACSAria (BD Biosciences) or SY3200 (Sony).

## qRT-PCR

RNA was extracted from naïve and cultured B cells using Trizol and RNeasy kit and Dnase treatment (Qiagen) and retro-transcribed to cDNA using random hexamers (Roche) and SuperScript II reverse transcriptase (Invitrogen). qRT-PCR was performed using Sybr Green in the 7900 Fast Real Time thermocycler (Applied Biosystems). The following primers were used: mouse- GAPDH (forward) 5'-TGA AGC AGG CAT CTG AGG G-3', (reverse) 5'-CGA AGG TGG AAG AGT GGG AG-3'; Total AID (forward) 5'- ACC TTC GCA ACA AGT CTG GCT -3', (reverse) 5'- AGC CTT GCG GTC TTC ACA GAA -3' and R26-AID (forward) 5'-ATT CGC CCT TGG GAT CCA-3', (reverse) 5'-AGC CAG ACT TGT TGC GAA GGT-3'. mRNA expression was normalized to GAPDH mRNA measured in triplicates in each sample. Relative quantification was performed using the $\Delta\Delta C_T$ method.

## Immunoblotting

Naïve and activated B cells were incubated on ice for 20 min in NP-40 lysis buffer in the presence of protease inhibitors (Roche) and lysates were cleared by centrifugation. Total protein was size-fractionated on SDS-PAGE 12% acrylamide-bisacrylamide gels and transferred to Protan nitrocellulose membrane (Whatman) in transfer buffer (0.19M glycine, 25 mM Tris base and 0.01% SDS) containing 20% methanol (90 min at 0.4A). Membranes were probed with anti-mouse-AID (1/25, eBioscience, 14-5959-82) and anti-mouse-tubulin (1/5000, Sigma-Aldrich, T9026) or anti-β-actin (1/40000, clone AC-74, Sigma A2228). Then, membranes were incubated with HRP-conjugated anti-rat (1/5000, Bethyl Laboratories, A110-

105P) and anti-mouse (1/10000, DAKO) antibodies, respectively and developed with Clarity™ Western ECL Substrate (Bio-Rad). Quantification of AID and actin/tubulin specific bands was performed with ImageJ software as integrated density values.

## IgH Sμ amplification and mutation analysis by PCR-Seq

For the analysis of mutations at IgH Sμ region, genomic DNA was isolated from GC and naïve B cells from immunized mice. Amplifications were performed in 4 independent reactions using the following primers:

(forward) 5'- AATGGATACCTCAGTGGTTTTTAATGGTGGGTTTA-3'; (reverse) 5'- GCGGCCCGGCTCATTCCAGTTCATTACAG-3'. Amplification reactions were carried in a final volume of 25 μl each, using 2.5 U Pfu Ultra HF DNA polymerase (Agilent) for 26 cycles (94°C for 30", 55°C for 30", 72°C for 60"). PCR products were purified and fragmented using a sonicator (Covaris), and libraries were prepared according to the manufacturer's instructions (NEBNext Ultra II DNA Library Prep; New England Biolabs). Sequencing was performed in a HiSeq 2500 platform (Illumina). Analysis was performed as previously described [30].

## Immunohistochemistry

Spleens and tumors were fixed with 10% neutral-buffered formalin and embedded in paraffin. 5 μM sections were stained with hematoxylin and eosin (Sigma) using standard protocols. For Ki67, Bcl6 and Pax5 immunohistochemistry, sodium citrate buffer (10mM pH 6, Sigma-Aldrich) was used for antigen retrieval. Endogenous peroxidases were blocked with 3% (v/v) $H_2O_2$ (Merck) in methanol (Sigma-Aldrich). For AID immunohistochemistry, EDTA buffer (1mM EDTA, 0,05% Tween-20 pH 8) was used for antigen retrieval and endogenous peroxidases were blocked with 0.3% $H_2O_2$ in water. Rabbit anti-Ki67 (Abcam, 1/100), goat anti-Pax5 (Santa Cruz Biotechnology, 1/2000), rat anti-AID (eBioscience, 1/50), and anti-Bcl-6 (IG91E, generously provided by Dr G Roncador, CNIO) were incubated overnight at 4°C. Biotinylated secondary antibodies were incubated 1 hour at room temperature. Biotinylated antibodies were detected with the ABC system (Vector Laboratories). Images were acquired with a Leica DM2500 microscope or digitized in a Hamamatsu NanoZoomer Scanner. Tumors were diagnosed by a pathologist. Two criteria were used to categorize tumors as SB lymphoma or LB lymphomas 1) tumoral growth pattern; SB lymphomas show a follicular pattern and LB lymphomas show a diffuse growth pattern and 2) cellular morphology; tumors with predominance of centrocyte morphology (small cleaved irregular nuclei and inconspicuous nucleoli) were classified as SB lymphomas and those with a majority of the large centroblasts with open vesicular nuclei and nucleoli attached to the nuclear membrane were classified as LB lymphoma.

## Whole Exome Sequencing (WES)

Whole exome sequencing was performed in 15 tumors and one healthy tissue (tail) as a non-tumoral control. In brief, genomic DNA was isolated by standard procedures, quantified in a fluorometer (Qubit; Invitrogen), fragmented in a sonicator (Covaris) to ∼200 nucleotide-long (mean size) fragments and purified using AMPure XP beads (Agencourt). Quality was assessed with the 2100 Bioanalyzer (Agilent). Then, fragment ends were repaired, adapters were ligated, and the resulting library was amplified and hybridized with a SureSelect Mouse All Exon kit library (Agilent) of RNA probes. DNA–RNA hybrids were then captured by magnetic bead selection. After indexing, libraries were paired-end sequenced in a HiSeq 2500 platform to produce 2x100bp reads. DNA capture, library preparation, and DNA sequencing were performed by the Genomics Unit at Centro Nacional de Investigaciones Cardiovasculares (CNIC).

**a. Identification of somatic variants from WES data.** Raw reads were demultiplexed and processed into *fastq* files by CASAVA (v1.8) and quality checked by FastQC (0.10.1). Sequencing adaptors were removed by cutadapt (1.7.1) (minlength = 30; minoverlap = 7) and the resulting reads were mapped to the mouse genome (GRCm38 v75 Feb 2014) using BWA-MEM (0.7.10-r789) (command line option -M to mark shorter split hits as secondary). Duplicates were marked by Picard tools (1.97). Variant calling was performed by means of GenomeAnalysisTK-3.5 (GATK3.5) and GATK Resource Bundle GRCm38 using default parameters and thresholds: 1) Aligned reads were realigned for known insertion/deletion events (Realigner TargetCreator and IndelRealigner tools) to minimize the number of mismatching bases and reduce false SNV calls; 2) base qualities were recalibrated (BaseRecalibrator tool) by a machine learning model to detect and correct systematic errors in base quality scores; 3) a custom Panel Of Normals (PON) was built; 4) somatic variants were called by MuTect2 (from GATK3.6); 5) Variants were functionally annotated with Variant Effect Predictor (VEP) version 84 (command line option—per-gene to output only the most severe functional consequence of the variant at the gene level) and filtered out to remove false positives (custom scripting). PON was built by combining the following variants: set1, intersection between germline (HaplotypeCaller) and somatic (MuTect2) variant calls in healthy tissue; set2, somatic variants (MuTect2) that have been called in all the 15 tumor samples and healthy tissue, including variants labeled as "NO PASS" by the caller in ≤2 of the samples; set3, somatic variants (MuTect2) that have been called in ≥13 tumor samples and healthy tissue and that have been labeled as "PASS" by the caller in all the samples. Filtering out of potential false positives was done by removing all variants that were called in healthy tissue, annotated in dbSNP (v142) or annotated in the Mouse Genomes Project (v5) for 129P2/OlaHsd, 129S1/SvImJ, 129S5SvEvBrd, BALB/cJ, C57BL/6NJ and CBA/J strains. Exome sequencing metrics included in S1 Table were calculated using Picard tools HsMetrics.

**b. Copy number variation analysis from WES data.** Copy number variation analysis was done using CopywriteR, an R package which implements a statistical method to estimate CNV from off-target reads in capture-based sequencing experiments. This approach overcomes most of the limitations and biases associated to CNV estimation from targeted sequencing data. Briefly, the software reads the alignment files, removes non-random off-target reads, calculates depth of coverage in predefined bins, corrects it by GC-content and mappability and generates a CNV profile. We used 20Kb bins and provided both tumor and healthy tissue (as a negative control) samples for the software to call CNV regions that exclusively occur in tumors.

**c. Translocation analysis.** Raw WES reads were processed and aligned to the mouse genome (GRCm38 v75 Feb 2014) as indicated in section "Identification of somatic variants from WES data". Translocations were identified using Manta1.6.0 with default parameters: 1) Execute configManta.py to configure the analysis:--exome parameter to indicate that reads come from whole exome sequencing;--normalBam and--tumorBam to indicate normal and tumor samples, respectively;--referenceFasta to indicate genome of reference used for the alignment;--runDir to indicate working directory; 2) Execute runWorkflow.py to run the analysis. Both configManta.py and runWorkflow.py are part of the standard distribution of Manta software. Translocations occurring in > = 2 tumors (same breakpoint coordinates) were considered as germinal and discarded from the analysis.

**d. Inference of clonal architecture of tumors and estimation of intratumor heterogeneity from WES data.** Intratumoral heterogeneity was estimated using the R package SciClone [61]. Variant allele frequencies (VAFs) and CNV information were used to infer clonal architecture of each of the 15 tumors analyzed. To ensure statistical robustness, only those variants supported by ≥ 100 reads were considered for the ITH analysis, as recommended by the

authors of the tool. Mutations with a VAF greater than 0.8 were excluded to filter out germline mutations. Variants belonging to the cluster with the highest VAF were considered as "clonal" and those belonging to clusters with a lower VAF were annotated as "subclonal".

**e. Identification and functional annotation of cancer associated genes from WES data.** Cancer associated genes were identified as those genes that bear at least one variant in one of the 15 tumors analyzed and that are annotated in IntOGen [65] as cancer drivers (gene set composed of 459 genes) or appear mutated in human diffuse large B cell lymphoma, Burkitt lymphoma or follicular lymphoma tumors [65–71] (lymphoma associated genes; gene set composed of 239 genes). Functional annotation of the 132 and 154 cancer-related genes identified in $R26^{+/+}Cd19^{Cre/+}Ung^{-/-}$ and $R26^{+/AID}Cd19^{Cre/+}Ung^{-/-}$ tumors was performed by Gene Ontology Biological process database. Enrichment tests were done using DAVID 6.8.

## Statistical analysis

Statistical analyses were performed with GraphPad version 6.01 or stats R package v3.1.1. Error bars in figures represent SD. Two-tailed Student's t test was applied to continuous data following a normal distribution (Shapiro-Wilk test) and two-tailed Fisher or $\chi^2$ test were used to assess differences between categorical variables. Kaplan-Meier survival curves were compared by log-rank Mantel-Cox test. Differences were considered statistically significant at $P \leq 0.05$.

## Supporting information

**S1 Fig. Mouse model for the conditional expression of AID in B cells. (A)** Representation of the cassette used to generate the $R26^{+/AID}Cd19^{+/Cre}$ mouse model. An AID-IRES-GFP construct preceded by a transcriptional STOP flanked by loxP sites was inserted within the *Rosa26* locus. $R26^{+/AID}$ mice were then crossed with $Cd19^{+/Cre}$ mice to generate $R26^{+/AID}Cd19^{+/Cre}$ mice. **(B)** Progressive *R26-AID* allele expression during B cell differentiation. Representative FACS analysis of GFP reporter expression in bone marrow and splenic B cells from $R26^{+/AID}Cd19^{+/Cre}$ (black empty line) and $R26^{+/+}Cd19^{+/Cre}$ (grey shade) mice. Bone marrow B cell subsets were gated as prepro (B220+CD19-), pro (B220+CD19+IgM-CD25-), pre (B220+CD19+IgM-CD25+), immature (B220+CD19+IgM+IgD-) and recirculating (B220+CD19+IgM+IgD+). Spleen B cells were gated as B220+. **(C)** B cells from $R26^{+/AID}Cd19^{+/Cre}$ mice develop normally. Left, representative FACS plots of prepro, pro, pre immature and recirculating B cells from $R26^{+/AID}Cd19^{+/Cre}$ and control $R26^{+/+}Cd19^{+/Cre}$ mice. Right, frequency quantification of each B cell subpopulation in $R26^{+/AID}Cd19^{+/Cre}$ (n = 9) and $R26^{+/+}Cd19^{+/Cre}$ mice (n = 8). Two-tailed t-test, error bars represent SD.
(TIF)

**S2 Fig. Mouse model for the conditional expression of AID in hematopoietic cells.** FACS analysis of GFP reporter expression in bone marrow, spleen and thymus cells from $R26^{+/AID}Vav^{+/cre}$ (black empty line) and $R26^{+/+}Vav^{+/cre}$ (grey shade). Bone marrow and spleen B cell subsets were gated as in S1B–S1C Fig. T cell subsets were gated as DN (CD4-CD8-), DN1 (CD4-CD8-CD44+CD25-), DP (CD4+CD8+), CD4+SP (CD4+CD8-), CD8+SP (CD4-CD8+) from thymus and naïve-T (B220-CD3+) from spleen.
(TIF)

**S3 Fig. Staining controls for Fig 4D.** Ki67 and AID immunohistochemistry of spleen sections from SRBC immunized *Aicda*$^{+/+}$ and *Aicda*$^{-/-}$ mice. Magnification is 5x inset is 40x. Scale bars are 500μm and 50μm for 5x and 40x images respectively.
(TIF)

**S4 Fig. Immunohistochemistry analysis (left) and immunophenotype (right) of 10 tumors from _R26_$^{+/+}$_Cd19_$^{+/cre}$_Ung_$^{-/-}$ mice.** Magnification for Ki67 and AID IHQ images is 40x. Scale bar is 50μm. Total live cells (left FACs panel) and B220+ gated cells (middle and right FACs panels) are shown.
(TIF)

**S5 Fig. Immunohistochemistry analysis (left) and immunophenotype (right) of 10 tumors from _R26_$^{+/AID}$_Cd19_$^{+/cre}$_Ung_$^{-/-}$ mice.** Magnification for Ki67 and AID IHQ images is 40x. Scale bar is 50μm. Total live cells (left FACs panel) and B220+ gated cells (middle and right FACs panels) are shown.
(TIF)

**S6 Fig. Exome sequencing of tumors. (A)** Total number of SNVs and INDELS identified in each of the 15 tumors analyzed. **(B)** Average proportion of SNVs and INDELs found in _R26_$^{+/+}$_Cd19_$^{+/Cre}$_Ung_$^{-/-}$ and _R26_$^{+/AID}$_Cd19_$^{+/Cre}$_Ung_$^{-/-}$ tumors. **(C)** CNV analysis of tumors from WES data. Heatmap of the genomic distribution of CNV alterations in _R26_$^{+/+}$_Cd19_$^{+/Cre}$_Ung_$^{-/-}$ and _R26_$^{+/AID}$_Cd19_$^{+/Cre}$_Ung_$^{-/-}$ tumors. Upper panel depicts number of CNV regions per tumor, with colors encoding copy number gain (red) or loss (grey). **(D)** Number of translocations identified in _R26_$^{+/+}$_Cd19_$^{+/Cre}$_Ung_$^{-/-}$ and _R26_$^{+/AID}$_Cd19_$^{+/Cre}$_Ung_$^{-/-}$ tumors by Manta analysis of WES data (two-tailed t-test, p = 0.852; tumor vs healthy tissue, see methods for details).
(TIF)

**S1 Table. Whole Exome Sequencing (WES) metrics of the 16 samples analyzed.**
(XLSX)

**S2 Table. Tumors samples analyzed by WES.**
(XLSX)

**S3 Table. Variants identified in AID KI UNG KO and AID WT UNG KO tumors.**
(XLSX)

**S4 Table. Traslocation analysis of AID KI UNG KO and AID WT UNG KO tumors.**
(XLSX)

**S5 Table. List of cancer-related genes mutated in AID KI UNG KO and/or AID WT UNG KO tumors.**
(XLSX)

**S1 Appendix. Mutation analysis of PCR-seq data at the IgH Sμ region from GC and naïve B cells.**
(XLSX)

## AcknowledgmentsAckowledgments

We thank all members of the B cell biology lab for scientific discussion, JM Ligos and the CNIC Cellomics Unit for flow cytometry assistance, CNIC Genomics Unit for assistance with whole genome sequencing and PCRseq experiments, CNIC Histology Unit for assistance with immunohistochemistry stainings and Carlos Torroja for helpful advice with whole exome sequencing data analysis.

## Author Contributions

**Conceptualization:** Pilar Delgado, Ángel F. Álvarez-Prado, Laura Belver, Virginia G. de Yébenes, Almudena R. Ramiro.

**Formal analysis:** Pilar Delgado, Ángel F. Álvarez-Prado, Ester Marina-Zárate, Isora V. Sernandez, Fátima Sanchez-Cabo, Marta Cañamero, Antonio de Molina, Laura Belver, Virginia G. de Yébenes, Almudena R. Ramiro.

**Funding acquisition:** Almudena R. Ramiro.

**Investigation:** Pilar Delgado, Ester Marina-Zárate, Isora V. Sernandez, Sonia M. Mur, Jorge de la Barrera, Laura Belver, Virginia G. de Yébenes.

**Methodology:** Pilar Delgado, Ángel F. Álvarez-Prado, Ester Marina-Zárate, Fátima SanchezCabo, Laura Belver, Virginia G. de Yébenes, Almudena R. Ramiro.

**Project administration:** Almudena R. Ramiro.

**Supervision:** Almudena R. Ramiro.

**Validation:** Pilar Delgado, Ángel F. Álvarez-Prado, Ester Marina-Zárate, Laura Belver.

**Writing – original draft:** Ángel F. Álvarez-Prado, Almudena R. Ramiro.

**Writing – review & editing:** Pilar Delgado, Ángel F. Álvarez-Prado, Ester Marina-Zárate, Virginia G. de Yébenes, Almudena R. Ramiro.

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
