## [Decision Letter · Decision Letter 0]

5 Aug 2020

Dear Dr Ramiro,

Thank you very much for submitting your Research Article entitled 'Interplay between UNG and AID governs intratumoral heterogeneity in mature B cell lymphoma' to PLOS Genetics. Your manuscript was fully evaluated at the editorial level and by independent peer reviewers. The reviewers appreciated the attention to an important problem, but raised some substantial concerns about the current manuscript. Based on the reviews, we will not be able to accept this version of the manuscript, but would be happy to receive a revised version. We cannot, of course, promise publication at that time.

Should you decide to revise the manuscript for further consideration here, your revisions should address the specific points made by each reviewer. In particular, it will be important to clarify aspects that are novel and those that are confirmatory of previous data and/or predictions.  We will also require a detailed list of your responses to the review comments and a description of the changes you have made in the manuscript.

If you decide to revise the manuscript for further consideration at PLOS Genetics, please aim to resubmit within the next 60 days, unless it will take extra time to address the concerns of the reviewers, in which case we would appreciate an expected resubmission date by email to plosgenetics@plos.org.

You can use this link to log into the system when you are ready to submit a revised version, having first consulted our Submission Checklist.

[LINK]

Please do not hesitate to contact us if you have any concerns or questions.

Yours sincerely,

David Weinstock

Guest Editor

PLOS Genetics

Gregory Barsh

Editor-in-Chief

PLOS Genetics

Reviewer's Responses to Questions

**Comments to the Authors:**

Reviewer #1: Delgado et al. showed that a conditional knock-in of AID at the Rosa26 locus, when turned on in B lineage, was insufficient to generate B cell lymphoma. However, if combined with an UNG deficiency, B cell transformation is readily detected. Whole exome sequencing shows elevated somatic hypermutation in these tumors with increased subclonal variability and a wider range of oncogenic variants. These data reaffirm the mutagenic (oncogenic) potential of AID-generated uracils and provide direct evidence that UNG safeguards genomic integrity (at least partly) against AID’s DNA mutator activity. The experimental design is sound and the conclusions are supported by the data. This study should be of interests to a wide range of audience.

A few points need to be addressed:

1. Although elevated mutations were seen in R26+/AID Cd19+/cre Ung-/- tumors, whether or not these mutations are drivers for tumorigenesis is not known. The authors need to at least acknowledge that.

2. Page 10, Ln 6. The mutation frequency analysis was restricted transitions at G/C pairs. The rationale for this restriction is not given. This seems unfair, especially without knowing whether tumorigenesis is driven only (or mostly) by these transition mutations (point #1).

Minor points:

1. There is a recent study showing that GC might not be the dominant site for CSR (PMID: 31375460), the authors might want to consider modifying the introduction (Page 3), unless they hold a different view of that study.

2. It has been reported that UNG protects against AID-induced telomere loss (PMID: 27697833). Although this may lead to B cell senescence instead of transformation, one cannot rule out the possibility that somehow telomere stress contributes to genome instability and oncogenesis. A brief discussion will be beneficial.

Reviewer #2: Dear Editor,

Thank you for the opportunity of reviewing the manuscript entitled: “Interplay between UNG and AID governs intratumoral heterogeneity in mature B

cell lymphoma” by Delgado et al.

The manuscript aims at addressing the contribution of AID in the process of lymphomagenesis with particular focus to the induction of SHM in lymphoma cells.

The Authors generated a new model of conditional supra-expression of AID selectively in B cells by knocking-in AID in the Rosa26 locus.

They observe lack of increase lymphoma formation with supra-expression of AID (probably achieving a lower over-expression than in other models of AID transgenesis) and absence of co-operation by p53 mutations. In contrast, a combined deficiency of UNG associated with AID supra-expression induces lymphoma formation in a subset of mice.

By characterizing the mutational patterns of these lymphoma, the Authors find that supra-expression of AID together with lack of UNG induces higher SHM rates in lymphomas, associated with higher subclonal heterogeneity of mutations.

Overall, the work is well presented and clearly written. Data are well organized, the design of the experiments is sound, the conclusions are supported by the data provided.

However, novelty is limited by the fact that most of the data are predictable from the current knowledge in the field, including the increased SHM rate when AID is coupled with UNG deficiency and the fact that UNG deficiency induces B cell lymphoma.

Major comments:

- the assumption that chromosomal translocation do not play a role in R26+/+-Cd19+/cre-Ung-/- and R26+/AID-Cd19+/cre-Ung-/- lymphomas has to be experimentally verified in this work. While it is established that UNG deficiency reduces chromosomal translocations driven by AID (as published in previous work by the corresponding Author), yet it is unclear whether translocations are completely abrogated or just reduced in UNG deficiency B cells, in particular when AID is overexpressed. The Authors should evaluated their tumors for the presence at least of c-myc translocations/amplifications.

- It is unclear which are the criteria the Authors use to distinguish lymphoma FL-like or DLBCL-like phenotype. This distinction needs to be clearly explained and supported by data, otherwise it could be misleading because typically DLBCL carry more mutations than FL. Since R26+/AID-Cd19+/cre-Ung-/- lymphoma show increased and broader SHM, the expectation is of an increased and not reduced incidence of DLBCL in these mice. Can the Authors explain?

Minor comments

- histology images should contain scale bars

- reference 10 and 54 are the same

Reviewer #3: Delgado et al generated mice that conditionally overexpress AID to test its contribution to B cell lymphoma development and evolution. AID o/e in the hematopoietic lineage, or from early B cell development is insufficient to affect mouse survival, even when combined with Tp53-/-. However, combined with Ung-/-, AID o/e from early B cell development results in increased lymphoma incidence compared to control Ung-/- mice. The lymphoma histopathology and GC marker expression profile in the Ung-/- R26AID are different from the Ung-/- but still mature B cell lymphoma. More importantly, WES showed that AID o/e in the absence of Ung increased heterogeneity in the tumoral clone with a clear AID mutational footprint. Mutations at genes that are often found mutated in human B cell lymphoma are also clonally mutated here, but they also analyze subclonal mutations, evidencing ongoing AID activity as a source of intratumor heterogeneity. They then focus on cancer-related genes, which are mutated in both mouse lines, but the AID o/e bears mutations in some additional genes.

This manuscript experimentally demonstrates that AID creates intratumor heterogeneity. While this might have been suspected, previous experimental systems could only show broad correlations of mutation signatures and did not analyze clonal and subclonal dynastic relationships. The overexpression system is in a way artificial but likely the only viable option to test this (see below). In addition, it allows them to better make the point that AID by itself is insufficient for lymphomagenesis. Since AID expression is well associated to poor outcome in human B cell neoplasms, the ITH demonstrated here acquires further relevance.

This is very good work, clearly written and technically sound. The data is solid, correctly analyzed, and very well discussed. I support publication. I have several question and comments that could be useful to address in the final manuscript, mostly to satisfy my curiosity.

The observation that AID overexpression does not lead to more CNV than endogenous AID seems worth emphasizing. This is a nice indication that most deletional / recombinogenic activity of AID is via UNG. If I am not mistaken, this has so far been inferred from the analysis of specific translocations, but not at the genomewide level. This could be emphasized in the discussion since it is important for their conclusion that point mutations by AID can accelerate lymphomagenesis.

This work compares tumors in Ung deficient mice that express two different doses of AID. The perfect but impossible control would have been similar tumors without AID, which just do not happen. This point could be made in the discussion for the sake of people not in the field.

I agree with their interpretation that moderate AID o/e may have partially contributed to avoid the negative regulation seen with transgenes and could explain the lack of tumors in wt or p53-/- background. Thus, it would be best to actually provide an estimate of how much more AID protein and activity is there in mature B cells in the R26AID compared to endogenous levels. AID protein could be quantified from the WB. The increase in IgG1 in vitro and Smu mutation data suggest ~2-fold but it may not represent activity off target. I guess the number of mutations at WRC could also be used as a proxy for off target activity, which seems to come about to 2-fold as well?

I am also curious about the level of AID in the final tumors. The IHC in Fig 4D would suggests some heterogeneity between R26AID tumors in either the level of AID observed in the lymphoma cells or proportion of cells expressing AID. Is there any way to quantify this?

In fig. 3, the authors show that mutation at the Smu is higher in UNG-/- cells when AID is o/e. However, endogenous AID barely increases mutations in UNG ko GC cells. This would suggest that UNG is not much of a break against endogenous AID mutagenic activity. Yet, Ung-/- tumors are said to show AID signature. It is not clear how this was determined. Does the total frequency of mutation differ between the two groups? Do mutations at C/G or at WRC happen at a higher than expected frequency in the R26+/+ Ung-/- tumors? It would be best to clarify this.

The previous point has implications for the etiology of B cell lymphoma found in R26AID but also in the control Ung-/- mice. Does AID contribute to lymphomagenesis in both? Both are mature but different, at least histologically. It would be useful to have the proportion of lymphoma in each line that were positive for Pax5, B220, GL7, etcl; to better assess how robust is the change in the type of lymphoma. Is there any indication that tumors from R26AID are more aggressive (i.e. more Ki67+?). Can they speculate on the cause? AID would be expressed very differently, over the whole development in R26AID versus only after activation in the controls. Maybe each lymphoma type derives from a different mature B cell type? Or is it just a reflection of mutations by the o/e AID targeting genes that are less frequently targeted by endogenous AID?

A further point raised by the previous comments is whether UNG is a tumor suppressor in humans. It is clearly so in the mouse and UNG deficiency is useful here to prove the point about ITH by SHM, but to my knowledge, UNG has not been singled out as a commonly mutated gene in human B cell lymphoma. The authors mention that UNG loss has been found in human lymphoma. It would be best to qualify this. How frequent is UNG loss of function?

**Have all data underlying the figures and results presented in the manuscript been provided?**

Reviewer #1: Yes

Reviewer #2: Yes

Reviewer #3: **No: **most numerical data is about the analysis of mutations from WES; so numbers may be covered by the data deposited. but there is no excel file with the numbers for plots shown in other figures (I apologize if I missed it)

PLOS authors have the option to publish the peer review history of their article (what does this mean?). If published, this will include your full peer review and any attached files.

Reviewer #1: No

Reviewer #2: No

Reviewer #3: No

---

## [Decision Letter · Decision Letter 1]

8 Nov 2020

Dear Dr Ramiro,

We are pleased to inform you that your manuscript entitled "Interplay between UNG and AID governs intratumoral heterogeneity in mature B cell lymphoma" has been editorially accepted for publication in PLOS Genetics. Congratulations!

Yours sincerely,

David Weinstock

Guest Editor

PLOS Genetics

Gregory Barsh

Editor-in-Chief

PLOS Genetics

Comments from the reviewers (if applicable):

Reviewer's Responses to Questions

**Comments to the Authors:**

Reviewer #1: My critics are adequately addressed.

Reviewer #2: Dear Editor,

the Authors have satisfactorily addressed all my comments as well as other Reviewer's questions.

The work is markedly improved and coveys important messages in a very elegant and carefully analyzed experimental model.

Reviewer #3: The authors have answered all my questions and addressed all my comments. I wish to thank them for their thorough answers.

**Have all data underlying the figures and results presented in the manuscript been provided?**

Reviewer #1: Yes

Reviewer #2: Yes

Reviewer #3: Yes

PLOS authors have the option to publish the peer review history of their article (what does this mean?). If published, this will include your full peer review and any attached files.

Reviewer #1: No

Reviewer #2: No

Reviewer #3: No

**Data Deposition**

http://datadryad.org/submit?journalID=pgenetics&manu=PGENETICS-D-20-00995R1

**Press Queries**

---

## [Editor Report · Acceptance letter]

10 Dec 2020

PGENETICS-D-20-00995R1 

Interplay between UNG and AID governs intratumoral heterogeneity in mature B cell lymphoma 

Dear Dr Ramiro, 

We are pleased to inform you that your manuscript entitled "Interplay between UNG and AID governs intratumoral heterogeneity in mature B cell lymphoma" has been formally accepted for publication in PLOS Genetics! Your manuscript is now with our production department and you will be notified of the publication date in due course.

With kind regards,

Livia Horvath

PLOS Genetics

On behalf of:
